# Neurogenic Bowel Dysfunction in Children and Adolescents

**DOI:** 10.3390/jcm10081669

**Published:** 2021-04-13

**Authors:** Giovanni Mosiello, Shaista Safder, David Marshall, Udo Rolle, Marc A. Benninga

**Affiliations:** 1Department of Surgery, Division of Urology, Bambino Gesù Pediatric and Research Hospital, 00165 Rome, Italy; 2College of Medicine, Center for Digestive, Health and Nutrition, Arnold Palmer Hospital for Children, Orlando, FL 32806, USA; shaista.safder@orlandohealth.com; 3Department of Pediatric Surgery and Pediatric Urology, Royal Belfast Hospital for Sick Children, Belfast BT97AB, UK; david.marshall@belfasttrust.hscni.net; 4Department of Pediatric Surgery and Pediatric Urology, Goethe-University Frankfurt, 60596 Frankfurt, Germany; udo.rolle@kgu.de; 5Department of Pediatric Gastroenterology, Hepatology and Nutrition, Emma Children’s Hospital, Amsterdam UMC, University of Amsterdam, 1105 AZ Amsterdam, The Netherlands; m.a.benninga@amsterdamumc.nl

**Keywords:** neurogenic bowel, bowel dysfunction, constipation, fecal incontinence, pediatric, children, adolescent, spina bifida, anorectal malformation, cerebral palsy

## Abstract

Neurogenic/neuropathic bowel dysfunction (NBD) is common in children who are affected by congenital and acquired neurological disease, and negatively impacts quality of life. In the past, NBD received less attention than neurogenic bladder, generally being considered only in spina bifida (the most common cause of pediatric NBD). Many methods of conservative and medical management of NBD are reported, including relatively recently Transanal Irrigation (TAI). Based on the literature and personal experience, an expert group (pediatric urologists/surgeons/gastroenterologists with specific experience in NBD) focused on NBD in children and adolescents. A statement document was created using a modified Delphi method. The range of causes of pediatric NBD are discussed in this paper. The various therapeutic approaches are presented to improve clinical management. The population of children and adolescents with NBD is increasing, due both to the higher survival rate and better diagnosis. While NBD is relatively predictable in producing either constipation or fecal incontinence, or both, its various effects on each patient will depend on a wide range of underlying causes and accompanying comorbidities. For this reason, management of NBD should be tailored individually with a combined multidisciplinary therapy appropriate for the status of the affected child and caregivers.

## 1. Introduction

Bowel dysfunction is reported in 0.7–29.6% of children and adolescents, and may be related to functional disorders or to congenital anatomical malformations or digestive tract and neurological causes [1,2,3]. Chronic constipation and fecal incontinence often coexist, sometimes with “overflow” diarrhea (where solid stool impacted higher up the rectum or colon only allows watery stool past it, which is then very difficult for even a neurologically intact anal sphincter to retain). This results in a frustrating situation for both patients and caregivers, especially in a neurogenic scenario, commonly defined as neurogenic or neuropathic bowel dysfunction (NBD) [4]. The term NBD implies autonomic and/or somatic denervation of the bowel. In pediatrics it is commonly thought of as being synonymous with spina bifida (SB) [5], without considering the wide range of other clinical conditions where NBD is present, such as in cerebral palsy, acquired brain and spinal cord injuries, transverse myelitis, etc. Globally, NBD in the pediatric population is still often not adequately considered or treated with the same standardized approach as neurogenic or neuropathic bladder dysfunction [6]. Yet addressing NBD is likely also to produce secondary benefits for the urinary tract, particularly improved functional bladder capacity (due to the rectum no longer compressing it, and reduction in the reflex detrusor overactivity that was promoted by rectal distension) and lower incidence of UTIs (thanks to improved bladder dynamics and the decrease in soiling of the perineum). Several approaches have been reported for managing NBD. However, it is common to observe many patients failing to respond to standard conservative and medical treatments such as dietary manipulation, manual evacuation, oral laxatives, suppositories, and/or enemas, with about half remaining fecally incontinent [7]. Before the introduction of a revised method of transanal irrigation (TAI) [8], many children were treated surgically with a Malone antegrade continence enema [7] or colostomy. TAI has transformed the management of NBD, and now is widely used in adults and children with SB [9,10,11,12]. Thanks to the improved survival of children with neurological conditions that were previously considered fatal, and better awareness and diagnostic techniques, the worldwide population of children and adolescents with NBD is growing. The aim of this paper, therefore, is to produce a statement regarding NBD in childhood and adolescent populations. We suggest to healthcare providers (HCPs) the clinical situations in childhood where NBD should be considered and investigated, and report recommendations for care, in order to improve the clinical outcome of management globally, as well as identifying issues for future research.

## 2. Methods

A group of specialists from different disciplines (pediatric gastroenterology, pediatric surgery, and pediatric urology) around the world (Europe and USA), with long-term experience in bowel management, was previously convened by Coloplast A/S, and was tasked to produce in 2017 best-practice recommendations for NBD management in pediatrics [13], an area of commercial interest for the company. A similar group decided to compile this follow-up report based on the current literature and personal experience, in order to offer a practical instrument for all HCPs involved in diagnosis and management of NBD in pediatrics, using a modified Delphi method [14]. Coloplast A/S, (Kokkedal, Denmark) sponsored the publication fee of this article through an educational grant, although its content was developed independently and was not in any way influenced by Coloplast A/S.

Three main topics were explored:The causes and pathophysiology of NBD in children and adolescentsThe conservative and medical management of NBD in children and adolescentsThe surgical management of NBD in children and adolescents.

Panelists selected a topic of interest, then literature data were selected and reviewed independently. The validity assessment of the literature data was performed independently. Each participant produced their own draft document in their area of special interest that were then combined and revised to produce a preliminary team consensus. All panelists next reviewed the preliminary document, offering their final opinions and revisions. Changes were made accordingly to obtain this final unanimously agreed paper. There were no critical points of discordance.

Literature research was obtained using PUBMED and Cochrane database using the following keywords as search terms: transanal irrigation, bowel management in children/adolescents, neurogenic bowel in children/adolescents, neurogenic bowel in pediatrics/young adults, fecal incontinence in children/adolescents, constipation in children/adolescents, bowel management in anorectal malformation, bowel management in spinal dysraphism, surgery for bowel management. All identified papers were screened for relevance based on title and abstract in the English language.

## 3. Results

### 3.1. The Causes and Pathophysiology of NBD in Children and Adolescents

#### 3.1.1. Causes

The causes and presentation of a neurogenic bowel dysfunction (NBD) in children and adolescents are different from adult forms. In most cases, pediatric NBD is caused by congenital problems such as spina bifida (SB). Acquired forms caused by trauma, infection, etc., are more similar to adult clinical pictures [15]. A range of etiologies are presented below, in approximate order of their pediatric relevance (determined by the prevalence of the cause in childhood and each cause’s propensity to cause NBD in children).

##### Myelodysplasia

Commonly known as SB, myelodysplasia describes incomplete closure of the vertebral column and malformation of the embryonic neural tube. This term includes a group of neural tube defects (NTDs) ranging from spina bifida occulta to meningocele to myelomeningocele and lipo-myelomeningocele. Myelomeningocele (MMC) is one of the most common birth defects of the spine and brain, potentially involving any level of the spinal column (lumbo-sacral 47%, lumbar 26%, sacral 20%, thoracic 5%, and cervical spine 2%) [16]. The neurological lesions produced by SB are variable and contingent on the neural elements that protrude within the sac. In myelomeningocele, the neural roots or segments of the spinal cord herniate through the incompletely closed vertebrae, and so are exposed to damage antenatally (and/or postnatally, until the sac is surgically closed). However, the bony vertebral level correlates poorly with the neurological lesions produced. Moreover, during childhood from birth to puberty, different growth rates between the vertebral bodies and the elongating spinal cord can introduce a dynamic factor to the neurological lesion. Furthermore, scar tissue congenitally surrounding the cord at the site of the MMC, and/or acquired following surgical closure of the MMC, can produce primary and secondary tethering of the cord, leading to a changing neurological picture during periods of rapid growth.

Associated hydrocephalus (with an Arnold–Chiari, or Chiari type-II, malformation) is seen in 85% of children with MMC, often requiring ventriculo-peritoneal shunting of excess cerebrospinal fluid to reduce the impact of its pressure on the brain.

Widespread mandatory fortification of dietary staples with folic acid, and voluntary ingestion of folic acid prior to conception and during the first trimester of pregnancy, have significantly reduced the incidence of MMC and other neural tube defects [17]. The vast majority of cases of MMC affect the lumbar spinal cord and sacral roots that innervate the bladder, distal colon, and their respective sphincters, so some degree of neurogenic bladder and bowel dysfunction is almost universal in this population. The incidence of urethro-vesical dysfunction in myelomeningocele is not absolutely known, but most studies suggest it is very high (>90%). Similarly, anorectal dysfunction is very common [5].

By contrast, in meningocele the meninges protrude through a vertebral canal defect, but the neural elements of the cord remain confined within the canal and so generally are not damaged either antenatally or postnatally.

In occult myelodysplasia or occult spinal dysraphism or closed SB, the bony lesions are not open, so most cases have no evident signs of neurological lesion. Its incidental diagnosis is rising due to increasing use of spinal x-rays, ultrasonography and magnetic resonance imaging. In up to 90% of affected individuals, inspection reveals a cutaneous stigma overlying the lower spine such as a dimple, skin tag, hairy patch, hemangioma or subdermal lipoma. In some, subtle alterations may subsequently be found in the toes and feet, with discrepancies in lower extremity muscle size and strength, or abnormal gait. Back pain and an absence of perineal sensation are common symptoms in adolescents [18]. In symptomatic cases, the incidence of lower urinary tract dysfunction (e.g., urinary tract infection or urinary incontinence) is high, which is more commonly recognized as abnormal than are symptoms of NBD. Occult lesions may also become manifest with tethering of the cord later in life. This can lead to changes in bowel, bladder, sexual and lower extremity function [19].

##### Sacral Agenesis

Sacral agenesis (SA), or Caudal Regression Syndrome, is another neural tube defect, involving complete or partial absence of the lowest five vertebrae. Urinary and/or fecal incontinence are commonly described and recognized when the child fails toilet training on time. A careful inspection may reveal flattened buttocks [20,21], but a full physical examination should also include palpation of the spine to the tip of the coccyx (to exclude a bony defect), as well as neurological examination of the lower limbs and gait. SA is commonly associated with an anorectal malformation.

##### Anorectal Malformation

Anorectal malformation (ARM, previously referred to by the narrower term imperforate anus) has an estimated incidence ranging between 1 in 2000 and 1 in 5000 live births. It may occur as an isolated lesion or in conjunction with other congenital malformations, where spinal cord pathology occurs in 38% of cases [22]. The VATER or VACTERL association is a group of commonly coexisting abnormalities including Vertebral, Anorectal malformation, Cardiac, Tracheo-Esophageal fistula, Renal and Limb anomalies. ARM has previously been classified as high, intermediate, or low depending on whether the blind-ending rectum terminates above, at, or below the levator ani muscle. In the past, imperforate anus repair for high lesions involving a perineal approach to pull the rectum through to the anal verge frequently resulted in a pudendal nerve injury. With the innovation of the posterior sagittal anorectoplasty (PSARP) surgical approach this complication has been eliminated [23]. However, reports of spinal magnetic resonance imaging (MRI) reveal a 35–50% incidence of distal spinal cord abnormalities in children with ARM [22]. A complete pre-operative evaluation is recommended in all patients in order to detect early any spinal cord or bony malformation that may produce autonomic dysfunction [24], such as a low-lying conus medullaris (terminating below the normal L2 level). This can be more easily achieved by ultrasound scan in the first three months of life, before the spinal window ossifies [25]. It is also thought that pelvic nerves, both sensory and motor, may be affected at the same critical stage of fetal development as the anorectal region and the pelvic floor musculature. Therefore, the degree of long-term neuropathic bowel dysfunction in ARM depends not only on the extent of the congenital defect itself (both the intestinal anomaly and the associated congenital neuropathy), but also on possible iatrogenic damage by surgery and/or by potential secondary neuromuscular impact from inadequate medical evacuation of the rectum during infancy and childhood.

##### Cerebral Palsy

Cerebral palsy (CP) is defined as a congenital neurological condition due to non-progressive injury (typically presumed anoxic) or malformation of the brain occurring in the fetal or perinatal period [26]. The incidence of CP is about 1.5 per 1000 births, making it the most common neurological condition encountered in pediatrics. CP encompasses a group of disorders of differing degrees of the development of movement and posture. Up to 90% of the children with CP suffer from constipation and 47% fecal incontinence, though most to a minor extent [27]. These effects arise due to deranged higher-level control of the bowel and/or sphincter rather than a primary intrinsic neuropathy of these structures. About half of individuals with CP are intellectually disabled [28,29], which affects what treatment modalities for NBD are appropriate.

##### Muscular Dystrophies and Mitochondrial Disorders

Congenital muscular dystrophies (MD) are a wide group of muscle disorders that present with very early onset of muscular weakness. Affected individuals report symptoms of both bladder and bowel dysfunction. The most common bowel complaint is constipation, which in X-linked Duchenne Muscular Dystrophy (DMD) can become life-threatening, but generally the most disabling is fecal incontinence [30,31]. One series reported that in 47% of boys with DMD undergoing a colonic transit test, the radiopaque markers were retained in the recto-sigmoid, suggesting functional pelvic outlet obstruction [30]. According to Lo Cascio and colleagues, there is a substantial risk in patients with Duchenne Muscular Dystrophy of altered gastrointestinal (GI) transport and possible sensory impairment, due to expression of dystrophin isoform DP116 in peripheral nerve tissue and autosomal homologues of DP116 in sensory ganglia [30]. Also implicated in impaired GI tract motility are alterations of the myenteric plexus associated with reduced myo-electrical slow-wave activity (as shown in mice models), along with a reduced availability of nitric oxide (NO), due to the lack of dystrophin acting as an anchor for NO-synthase [30]. These direct effects on GI transit time are not helped by a decreased ability to strain voluntarily, and sometimes further exacerbated by the developmental delay seen in some forms of mitochondrial disorder.

Loss of the alpha-dystroglycan-laminin interaction, due to defective glycosylation of alpha-dystroglycan, underlies a group of congenital muscular dystrophies often associated with brain malformations, referred to as dystroglycanopathies, where NBD is reported [31]. Mitochondrial neurogastrointestinal encephalomyopathy (MNGIE) is frequently associated with chronic intestinal pseudo-obstruction. The pathophysiology resulting in impaired peristalsis and propulsion of intestinal contents relates to disturbed neuromuscular coordination due to myopathy (affecting intestinal contraction), neuropathy (affecting coordination of enteric reflexes), or mesenchymopathy (related to abnormalities of the interstitial cells of Cajal). Furthermore, mitochondrial abnormalities observed in MNGIE may contribute to disturbed homeostasis of gut microbiota, which may in turn be involved in the manifestation of gastrointestinal dysmotility seen in MNGIE [32].

Wolfram syndrome is a neuro-degenerative disorder characterized by childhood-onset diabetes mellitus, optic nerve atrophy, diabetes insipidus, hearing impairment, and commonly bowel and bladder dysfunction [33].

In all these muscle disorders, muscular dystrophies and mitochondrial cytopathy, NBD, and lower urinary tract symptoms can change over time with disease progression, so careful follow-up is required.

##### Acquired Brain Injury

Acquired brain injury (ABI) refers to a brain insult sustained after a period of normal development. ABI in children and adolescents is relatively common, with a heterogeneous group of underlying causes including vascular, oncological, and trauma (e.g., road traffic or sport). ABI represents the leading cause of death and neurologic disability in children after infancy. In the aftermath of more serious physical injuries, bladder and bowel dysfunction are often considered of secondary importance and managed with continence pads only until a delayed diagnosis and definitive management [34]. However, urinary retention and constipation often produce long-term urinary and fecal incontinence. Today survivors of ABI are increasing and comprise a large proportion of the work of a neurorehabilitation department. Functional impairments (motor, behavioral, educational, and cognitive) are common and can endure for life.

##### Acquired Pelvic Injury

NBD can occur from damage to the nerves innervating the pelvic organs, anywhere in the course of these nerves from the cauda equina, the spinal nerve roots, the sacral plexus, or to the various individual nerves that arise from the plexus. Most injuries to these nerves are iatrogenic, but rarely can occur as a result of high-impact trauma. Any pelvic surgery in infants and children for anorectal malformation (to mobilize the blind-ending rectum from the urinary tract, if necessary, and then to open it and bring it to the center of the anal sphincter complex) or Hirschsprung’s disease (to pull through normally ganglionated bowel to the anus) [35,36], neuroblastoma, ganglioneuroma, sacrococcygeal teratoma, and aorto-iliac surgery is theoretically able to damage the pelvic parasympathetic nerves to the rectum, anus, bladder, and genitalia. Additionally, pelvic irradiation can cause damage to adjacent nerve fibers, resulting in altered function, as can certain cytotoxic drugs [35]. Bowel, voiding, and erectile dysfunction can result. Iatrogenic fecal incontinence can also be caused by sphincter damage caused during childbirth (including in post-pubertal teenagers), and surgery for anorectal problems such as trauma, fistulae, and abscesses. Vaginal delivery can damage not only the anal sphincteric muscle, but also the neurons innervating the anal sphincter, especially in younger individuals [37].

##### Acquired Spinal Cord Injury

The relative flexibility of childhood tissues compared to adults confers a degree of protection against traumatic spinal cord injury (SCI). However, voluntary control of defecation requires rectal sensation, peristalsis, and adequate anorectal sphincter function and coordination. Neurological defects in patients with spinal lesions may affect one or more of these components, resulting in various types of defecation disorders: fecal incontinence, chronic constipation, or both [38].

NBD is common among pediatric patients with acquired SCI [39]. According to electromyography (EMG) of the external anal sphincter, 25–33% had bilateral or unilateral muscle action abnormalities during defecation, and 88.5% showed pelvic floor dysfunction. The mean rectal volume to generate the first sensation was significantly higher in SCI patients.

The level of spinal injury dictates two distinct clinical patterns of NBD [40], although both feature constipation. Injuries above the conus medullaris result in an upper motor neuron pattern of hyperreflexic bowel where inhibitory input is lost; this is characterized by increased colonic and anal sphincter tone, often resulting in stool retention and constipation. In upper motor neuron lesions, there is preservation of reflex coordination (such as the gastrocolic response), hypertonia (with consequent reduced rectal compliance), and hyperreflexia distal to the splenic flexure (resulting in reflex defecation and incontinence). In contrast, injuries at the conus medullaris or cauda equina result in a lower motor neuron pattern of areflexic bowel, with loss of centrally mediated motor activity leading to slow bowel transit and an atonic external anal sphincter; these patients may experience constipation yet also a significant risk of fecal incontinence. Lesions within the conus or in the cauda equina (where excitatory sacral parasympathetic supply is lost) are associated with rectal hypotonia and hyporeflexia, predisposing to impaction and overflow incontinence. The degree of symptoms also depends on the grade of injury: complete SCI has been shown to result in the most severe form of NBD with loss of voluntary control of the external anal sphincter too [41].

##### Down’s Syndrome

The secondary effects of Down’s Syndrome (DS) on the bowel and bladder have, until recently, often been dismissed as an inevitable reflection of the severely disabling primary condition, or simply behavioral. It is now increasingly recognized that DS is associated with significant lower urinary tract symptoms (LUTS) in children, which can even require surgical intervention [42,43]. It is believed this typically results from the child’s inability to relax their pelvic floor appropriately, leading to voiding dysfunction, fecal impaction, and secondary overflow incontinence. This bladder and bowel dysfunction in DS is undoubtedly multifactorial, but is often regarded as non-neurogenic neurogenic (as in Hinman bladder). However, the underlying neurological condition means it may have at least a neurogenic etiological component.

##### Autism

Similarly, it has historically been commonly assumed that inability to toilet-train is an unavoidable consequence of autism and its behavioral traits. Again, it is thought this bladder and bowel dysfunction is typically due to the child’s over-reliance on their pelvic floor muscles, and this often persists into adulthood [44]. While the cause of this dysfunction is presumably multifactorial, the underlying neurological condition suggests it is at least partially neurogenic.

##### Transverse Myelitis

Transverse myelitis (TM) is a rare immune-mediated process leading to neural injury in the spinal cord. TM is reported to have an incidence of 1–4 new cases per million, with bi-modal peaks between the ages of 10–19 years and 30–39 years, and is commonly para-infectious [45]. TM can clinically be divided into acute or sub-acute, affecting motor, sensory and/or autonomic nerves, so presentation can be varied with weakness, sensory alterations and, virtually always, autonomic dysfunction of storage and emptying of the bladder and bowel [46,47]. Approximately 20% of cases of acute TM occur in children, in whom one of the most common initial symptoms is pain (60%). Other common symptoms in children include motor deficits, numbness, ataxic gait, and loss of bowel or bladder control. Constipation can be severe and may present in children as increased irritability with fullness in the left lower quadrant. Long-term autonomic sphincter dysfunction is reported in different series in between 22–80% of children [48].

##### Guillain–Barré Syndrome

Guillain–Barré syndrome (GBS) has become the most common cause for acute, flaccid paralysis in many parts of the world [49]. It presents as a rapidly progressing ascending areflexic motor paralysis, with or without sensory and autonomic dysfunction. The initial acute progressive phase, which generally reaches a nadir within four weeks, is typically followed by a plateau phase and finally a recovery phase.

GBS affects children and adults of all ages and both sexes. Bowel dysfunction, seen in up to 15% of patients, occurs much less commonly than cardiovascular or limb dysfunction [50,51], so it is not often encountered by pediatric surgeons.

##### Cauda Equina Syndrome

The spinal cord terminates at the lower border of the L1 vertebra, and the nerve roots of L2 to S4 below form a tightly packed bundle, the so-called cauda equina (“horse’s tail”). Damage to these nerve roots produces a characteristic pattern of symptoms called the cauda equina syndrome (CES), where the predominant findings are bladder, bowel, and sexual dysfunction along with sensory loss of the perineum [52,53].

Central lumbar disc prolapse is rare in childhood, but can compress sacral nerve fibers to and from the bladder, the large bowel, the anal and urethral sphincters, and the pelvic floor, producing low-back pain, bilateral sciatica, saddle anesthesia, urinary retention, and constipation. Other causes include trauma, tumor, spinal canal stenosis, spinal AV malformation, and iatrogenic (during spinal surgery or spinal anesthesia) [54].

##### Multiple Sclerosis

Multiple sclerosis (MS) is the most common progressive neurological disorder in young people, with a mean age at onset of 30 years, and a prevalence of 40–220 cases per 100,000 people in Europe [55], with similar rates in North America [56]. The incidence of pediatric onset of MS is low at 0.3–0.9/100,000. The prevalence of pediatric MS is 5–10% of all cases of MS [57,58].

Bowel effects are common in patients who have MS and can have a significant impact on quality of life (QoL) [59]. Symptoms were found in 45 to 68% of cases and can be defined as “retentive” (constipation, seen in 31% to 54% of patients) or “irritant” (including diarrhea and false urges to defecate, with a prevalence of 6–20%) [60]. Data reported mainly refer to adult populations, although teenagers are included.

##### Acute Disseminated Encephalomyelitis and Meningitis-Retention Syndrome

Acute disseminated encephalomyelitis (ADEM), also known as postinfectious encephalomyelitis, is a rare demyelinating disorder of the central nervous system. It follows an exanthematous infection or, occasionally, vaccination [61], suggesting a para-infectious or autoimmune origin [62]. ADEM is characterized by an inflammatory reaction and demyelination in the brain and spinal cord. Lesions observed on MRI of the brain are usually confined to the white matter. Lesions in the spinal cord involving the conus are also seen [63]. The early symptoms of ADEM are similar to an acute relapse of multiple sclerosis (MS) and can cause diagnostic confusion in adolescents and young adults. Presence of symptoms such as fever and headache, along with a combination of signs of encephalitis (disturbance of consciousness, epilepsy, and hemi-paresis) and of myelitis (sensory disturbance below the level of the lesion, spastic paraplegia), may help differentiate this condition. Bowel and lower urinary tract dysfunction are common in ADEM [61,62,63,64].

Meningitis retention syndrome (MRS), a rare and very mild form of ADEM, has been recognized as a specific condition in its own right [63]. The frequency of NBD in pediatric MRS is not clear since it has a benign and self-remitting course (with a duration of 2–10 weeks). For the same reason, the effectiveness of immune treatments (steroid therapy) remains unclear, although such treatments may shorten the duration.

##### Spinal Canal Stenosis

While rarely arising in childhood, patients with spinal canal stenosis (SCS) may present with bladder and/or bowel involvement. To demonstrate narrowing of the lumbar canal with compression of the cauda equina by bony or soft tissue, CT or MRI is often recommended [65].

About half of the patients with intractable leg pain in spinal canal stenosis also have bladder and bowel symptoms from effects on the cauda equina. Schkrohowsky et al. [66] reported that one-third of patients with achondroplasia developed SCS, especially at the lumbar level. SCS is reported in Klippel–Feil and other syndromes [67]. Signs and symptoms of compressive neuropathy of multiple lumbar and sacral roots is an indication for surgical decompression, which usually improves the condition.

##### Other Rare Pediatric Neurological Disease

Motor neuron diseases or disorders (MNDs) are rare neurological conditions affecting the anterior horn cells of the motor neurons that control voluntary skeletal muscle activity (i.e., the external anal sphincter rather than the bowel itself, although they can also indirectly affect bowel function due to weakened abdominal musculature and immobility). MNDs are associated with a very poor prognosis since they are often progressive and currently no known cure is available (treatment being limited to symptomatic relief and supporting primary vital functions like breathing and feeding). MNDs can be classified according to the part of the body affected and the pattern of nerve involvement (upper or lower motor neurons, or both), and include spinal muscular atrophy (SMA), amyotrophic lateral sclerosis (ALS), progressive muscular atrophy (PMA), progressive bulbar palsy (PBP), and primary lateral sclerosis (PLS). While most of these are adult conditions, SMA is inherited and usually becomes symptomatic in childhood, with the earlier presentations seen in the more severe forms. Werdnig–Hoffmann disease is the most common and most severe form of SMA, type 1, whereas SMA-2 is an intermediate form compared to SMA-3; the mildest form, SMA-4, usually presents in early adulthood. The poor prognosis in MNDs such as SMA-1, means that NBD, although usually present, is often under-recognized [68].

In the most common disorder of the neuromuscular junction, myasthenia gravis, intestinal pseudo obstruction is reported [69].

X-linked adrenoleukodystrophy (ALD) is a neurometabolic disorder caused by mutations in the ABCD1 gene resulting in a defect in peroxisomal degradation of very long-chain fatty acids (VLCFAs) with their accumulation in plasma and tissues, affecting both spinal cord and brain. Bowel dysfunction and lower urinary tract symptoms have been described [70].

Menkes disease is an X-linked recessive disorder of copper metabolism due to a mutation in the ATP7A gene that leads to copper deficiency, culminating in a severe progressive neurodegenerative disease including bowel and bladder dysfunction [71].

In theory, any congenital or acquired medical condition that affects neurological and/or cognitive development and behavior can also produce secondary effects on the bowel (and bladder) of a child or adolescent.

##### Summary

NBD is commonly experienced in children and adolescents, often associated with neurogenic dysfunction of the bladder. In some neurological diseases, NBD has been more fully evaluated and early management is generally instituted. However, it is often missed, neglected, or undertreated in other conditions, both rare and common. Therefore, NBD must be considered in all children with special needs due to common congenital conditions like cerebral palsy and Down’s Syndrome, as well as all forms of rare acquired neurological damage such as post-trauma, Guillain–Barré syndrome, transverse myelitis, etc. Future research must address these clinical situations in order to define tailored diagnostic pathways and management.

#### 3.1.2. Pathophysiology

The term ‘neurogenic bowel’ encompasses the manifestations of bowel dysfunction resulting from sensory and/or motor disturbances due to central neurological disease or damage [72].

The gastrointestinal tract has a complex control that relies on coordinated interaction between muscular contractions and neuronal impulses [73]. Constipation and/or fecal incontinence occur when there is a problem with the normal bowel functioning; this could be for a variety of reasons. The usual defecation pathway involves contractions of the colon to help mix the contents, absorb water, and propel the contents along the intestine. This results in the feces moving through the colon to the rectum [74]. The presence of stool in the rectum causes a reflex relaxation of the internal anal sphincter (rectoanal inhibitory reflex, RAIR), allowing the contents of the rectum to move into the anal canal. This produces the conscious feeling of the need to defecate. At a socially suitable time, the brain sends nerve signals causing the voluntary external anal sphincter and puborectalis muscles to relax and this allows defecation to take place [15,74].

There are two main types of nervous system within the lower gastrointestinal (GI) tract: the intrinsic enteric nervous system (located within the wall of the gut) and the extrinsic nervous system (comprising sympathetic and parasympathetic innervation) [73]. The intrinsic enteric nervous system controls gut motility directly, whereas the extrinsic nerve pathways influence gut contractility indirectly by modifying this intrinsic enteric response [73]. In almost all cases of neurogenic bowel dysfunction, it is the extrinsic nervous supply that is affected while the intrinsic enteric nervous supply remains intact.

Defecation involves conscious and subconscious processes and, when the extrinsic nervous system is damaged, either of these can be affected. Conscious processes are controlled by the somatic nervous system; these are voluntary movements, for example the contraction of the striated muscle of the external anal sphincter is instructed by the brain, which activates the neurons innervating this muscle [75,76]. Subconscious processes are controlled by the autonomic nervous system; these are involuntary movements such as contraction of the smooth muscle of the colon or the internal anal sphincter. The autonomic nervous system also provides sensory information; this could be about the degree of distension within the colon or rectum [75,76].

Neurogenic bowel dysfunction (NBD) is generally related to spinal cord lesions in pediatric patients, mainly represented by open or closed neural tube defects (spina bifida) resulting from antenatal developmental neurological events.

Patients with spinal cord lesions, either congenital or acquired, have an anatomically intact rectal ampulla, anal canal, and sphincter but experience constipation and/or incontinence due to damage of their enteric nervous system, reduced sensation, and limited mobility. In these children, anal squeeze pressure, anorectal sensitivity, and anal resting pressure may also be impaired, while rectal compliance may be reduced due to hyperreactivity of the rectum [5,77], impacting colorectal motility, transit time and bowel emptying, which often leads to constipation, fecal incontinence, or a combination of both.

Bowel dysfunction occurs in children with spina bifida because, while their spinal rectoanal inhibitory reflex (RAIR) is generally maintained, their defecation urge is not present. When the internal sphincter relaxes, bowel soiling occurs. Constipation results from an increased colonic transit time and a lack of sphincter relaxation with rectal distension [77,78]. Additional factors leading to bowel dysfunction in children with spinal cord issues are a general decrease in activity and, depending on the level of the spinal lesion, abdominal muscle weakness resulting in a reduced ability to push out stool [79]. Most children develop constipation, typically passing frequent, small, and hard stools.

##### Impact of Anatomical Location of Nerve Damage

Damage to the spinal cord or brain can interrupt neural pathways. The location and severity of such damage are the key factors in determining colorectal function and the nature and extent of subsequent symptoms. However, it should be kept in mind that symptoms are not always easy to determine and can change with time. For instance, in spinal cord injury, the precise level of injury is often not clear during the early stages due to spinal shock, which can last up to six weeks. Moreover, the nervous system, being a complex entity, does not always present a fixed clinical pattern even in the same disease or trauma patterns.

Broadly, neurogenic bowel symptoms are divided into two patterns depending upon the level of disease or injury in relation to the conus medullaris:

1. Supraconal disorder—“upper motor neuron bowel syndrome” or “hyperreflexic bowel”, or “spastic bowel”

This pattern is seen in patients who have disease/injury above the conus medullaris and involves loss of supraspinal inhibitory input resulting in hypertonia of the colorectum. The increase in tone of the colonic wall, pelvic floor, and anus results in reduced colonic compliance, overactive segmental peristalsis, and underactive propulsive peristalsis [80,81,82]. As the peristaltic and haustral movements become less effective, the transit slows down throughout the colon [75,76]. The spastic constricted state of the external anal sphincter (EAS) worsens the situation further by causing retention of stool. The combination of these physiological responses to supraconal injury makes constipation the dominant gut symptom. When the anal sphincter cannot be voluntarily relaxed, signals between the colon and the brain become disrupted: the reflex that triggers a bowel movement still works, but the child may not feel it coming, resulting in a sudden unplanned passage of stool whenever the rectum is full. These disorders are characterized by high resting anal tone, anal/anocutaneous reflex present (reflex contraction of anus in response to stroking of perianal skin), and bulbospongiosus/bulbocavernosus reflex present (reflex contraction of anus in response to squeezing the glans penis or clitoris).

2. Infraconal disorder—“lower motor neuron type” or “areflexic bowel”

A flaccid bowel may follow a lower spinal cord injury. Infraconal lesions are a consequence of disruption of autonomic motor nerves due to damage to parasympathetic cell bodies in the conus medullaris or their axons in the cauda equina. This is characterized by loss of colorectal tone and reduced amplitude of rectoanal inhibitory reflex (RAIR), resulting in a cyclical pattern of insensate rectal filling and progressive rectal distension eventually leading to fecal incontinence. Furthermore, the incontinence is not helped by a reduction in resting and squeeze anal pressures due to flaccid anal sphincters and laxity of pelvic floor muscles which allows excessive descent of pelvic contents, reducing the anorectal angle and opening the rectal lumen [82]. In a flaccid bowel situation, there is reduced movement in the colon, less peristalsis, and the anal sphincter is more relaxed than normal. This can lead to constipation with frequent leaking of stool. Typically, these patients have no or low resting anal tone, and absence of the anal/anocutaneous and bulbospongiosus/bulbocavernosus reflexes.

##### Tools for Assessment of NBD

Important questions must first be asked during a thorough medical history regarding current stool frequency, consistency, and amounts. It is helpful to use a bowel diary to record the time(s) of the day when a bowel movement occurs and the presence of awareness or urge [83,84,85,86,87]. An accurate history should be obtained of both facilitators and barriers to success in any bowel management programs that have previously been attempted. Medications should be recorded, especially those that have the intended consequence or known side-effects of either constipation (since anticholinergics are commonly used for associated neurogenic bladder, or constipating-agents may deliberately have been used in an attempt to reduce soiling) or diarrhea. Indeed, in a situation of “overflow” diarrhea, sometimes antidiarrheal medication is unwittingly prescribed, which obviously only compounds the problem. Any previous surgery (especially abdominal and perineal) should be documented.

A full physical examination should include abdominal palpation for evidence of fecal loading, which would suggest that chronic constipation may be the cause of (overflow) fecal incontinence. Palpation is also used to assess for sensation, discomfort, tenderness, abdominal muscle tone and non-fecal masses. Percussion and auscultation of bowel sounds can suggest constipation, obstruction or pseudo obstruction. The perianal region should next be inspected for soiling, dermatitis, anal fissures, patulous anus, anal prolapse, or external hemorrhoids (although the latter are quite rare in children). Assessment should be made of perianal sensation and the corresponding reflex response of the anal sphincter. In obese individuals, fecal loading can be more accurately determined by judicious rectal examination, which can additionally provide evidence of the patient’s ability to produce voluntary contraction of the EAS and puborectalis muscles [88].

Various diagnostic tests can supplement the above clinical findings. In children with obesity or distorted body habitus (e.g., due to scoliosis associated with SB), abdominal x-ray can confirm fecal loading, although this does not always correlate well with symptoms [89], but it can provide useful evidence to convince skeptical parents or caregivers. Colonic transit time can be estimated by means of an abdominal x-ray taken a specified time after the child has swallowed small radio-opaque markers [90]. Anorectal manometry is a useful test to measure anorectal function and define NBD [91]. An endo-anal ultrasound can identify an external or internal anal sphincter defect, and barium enema or dynamic magnetic resonance (MR) proctogram can diagnose paradoxical sphincter contractions. Electromyography can test the electrical activity of the muscles around the anus and rectum. MRI or CT scan of brain and/or spinal cord may also be helpful in defining NBD. If clinical or radiological assessment raises a concern, it is important to exclude the rare possibility of a colonic stricture, if necessary by colonoscopy, before proceeding to any surgical treatment for symptoms that have been assumed to be caused by NBD.

### 3.2. The Conservative and Pharmacological Management of NBD in Children and Adolescents

Since NBD interferes with the normal voluntary control of defecation, the aim of all bowel management strategies is to allow emptying of as much as possible of the colon in the bathroom at a socially convenient time for the patient (and family), so that there is little or no possibility of fecal incontinence or constipation occurring whenever school, work, sport and hobbies, social activities, travel, or sleep preclude visiting the toilet at short notice. This target should be delivered with the minimum of time, fuss, discomfort/pain, side-effects, and expense. How exactly that is best achieved varies from child to child (and family), so treatment must be individualized and regularly reviewed to ensure it continues to meet this objective as the child grows and the degree of NBD perhaps alters with time.

#### 3.2.1. Starting a Bowel Management Program

The goal of establishing or maintaining a bowel management routine is to prevent constipation, optimize continence, maintain skin integrity [92], and maximize independence. When deciding on a treatment plan, it is essential first to undertake a thorough medical history and clinical examination (as outlined in Section Tools for Assessment of NBD above). This information will help providers to recommend a program most likely to succeed in the long term.

It is vital to establish from the offset whether an individual has a hyperreflexic or areflexic bowel, to help tailor their management accordingly. Patients with a hyperreflexic bowel have an intact reflex arc between the spinal cord and colon/anorectum and, as such, stimulation of the rectum (chemically or mechanically) results in evacuation of any rectal stool. The aim in hyperreflexic bowel is to attain a relatively soft stool consistency to encourage evacuation. In these patients, stool softeners and stimulant laxatives with mechanical stimulation of the anorectum may provide relief of stool.

On the other hand, individuals with areflexic bowel may require abdominal muscle exercises, gentle Valsalva maneuvers, and/or manual evacuation of stool. In these patients, who have low resting anal sphincter tone, more formed stool can help reduce incontinence episodes, so overuse of stool softeners and stimulant laxatives should generally be avoided.

In those with a neurological level at T_6_ or above, any treatment that produces rapid emptying of the rectum carries a threat of precipitating life-threatening autonomic dysreflexia [93]. At-risk patients or caregivers must be made aware of this danger and issued with advice on and supplies of appropriate emergency rescue strategies (such as nifedipine).

As previously proposed by this group (see modification in Figure 1), interventions should generally be considered in a stepwise manner, with the aim of finding the least invasive intervention that balances stool consistency and frequency, thus optimizing continence [13]. Treatments should be implemented for a minimum of two weeks consistently before considering altering the program further.

Tailoring treatment to the individual, considering whether they have upper or lower motor neuron bowel dysfunction, is important in the success of the bowel program [83,86]. In working with school-age children, consideration should also be given to the use of school staff to aid in tracking. The school nurse plays a vital role in assisting the child to reach their educational goals at the same time as managing their health concerns [81].

#### 3.2.2. The Pediatric Neurogenic Bowel Dysfunction Score

The Pediatric Neurogenic Bowel Dysfunction Score (PNBDS) is used widely by healthcare professionals managing children and adolescents with NBD. It is a validated standardized symptom-based measure of bowel function in patients who have neurogenic bowel. Such a scoring system was originally intended for use among adult patients with spinal cord injury and other neurological disorders and was initially validated in patients from 8 to 88 years old [94]. Thereafter, it was validated in the pediatric population ranging from 6 to 18 years old [95]. The PNBDS is derived from a 15-item questionnaire, covering bowel frequency, bowel continence, independence with bowel management, and impact on QoL of bowel symptoms and treatment. Scores are weighted based on QoL and can range from 0 to 41. A score <8 is considered to represent no bowel dysfunction, while higher scores are indicative of more severe NBD. Prior research has shown good measures of validity and reliability, making it useful as a monitoring tool to evaluate the efficacy of current bowel management regimens.

#### 3.2.3. Conservative Treatments

##### Dietary Patterns, Particularly Fiber

Changing diet to include higher fiber content is usually recommended as a first step in a bowel management program. For simplicity and safety, recommended minimal daily fiber intake (g/day) for children and adolescents from 3 to 20 years of age is calculated by the formula: age plus 5 g (e.g., 8 g/d at age 3 years, 15 g/d at age 10 years, and 25 g/d at age 20), and thereafter following adult guidelines of 25 to 35 g/d [96]. A well-balanced diet should be encourages, which includes fruits, vegetables, and plenty of water, and constipating foods such as cheese and white rice should be limited. Fiber supplements, which are often recommended for managing constipation for people with neurotypical bowel innervation, can cause constipation and discomfort for those with NBD and are not routinely recommended. However, it is important to understand the difference between soluble and insoluble fiber. Soluble fiber is hydrophilic; by attracting water, it removes excess fluid from the feces, making the stool more formed and decreasing liquid stool. Insoluble fiber, on the other hand, does not dissolve in water, so it stays intact as it moves through the digestive system, adding substance to the stool and so acts as a bulk-forming laxative. Soluble fiber includes plant pectins and gums commonly found in foods like lentils, peas, oats, barley, apples, and citrus foods. Insoluble fiber includes plant cellulose and hemicellulose including whole wheat or bran products, green beans, potatoes, cauliflower, and nuts. The fluid/fiber ratio is also important: inadequate fluid intake with the fiber can make constipation worse. A systematic review looking at non-neurogenic chronic idiopathic constipation concluded that, although few studies have shown benefit from using soluble fiber in this patient group, the evidence for using insoluble fiber is conflicting [97].

Similar results were reported by Markland et al. in their review of more than 10,000 adults, where they found a beneficial effect of increasing intake of fluid but not of fiber or exercise in managing constipation [98]. Looking specifically at individuals with NBD, a case series of 11 adults with SCI reported an increase in colonic transit time (i.e., constipation), rather than an improvement, with the use of insoluble fiber [99].

Consumption of a very high-fiber diet without proper advice on fluid intake may worsen constipation symptoms in certain patients who are fluid-sensitive. An individualized approach should be used with the use of insoluble fiber as a bulk-forming agent and factoring in fluid intake to optimize stool consistency [100,101,102,103].

##### Oral Fluid Intake

Good hydration is an important component of successful bowel management. Adequate fluid intake optimizes the effect of osmotic laxatives and fiber and is also necessary for bowel health overall. Fiber absorbs large amounts of water in the intestine, so a high-fiber diet can cause constipation if plenty of fluids are not also taken. Based on a normal child’s weight, their recommended daily fluid intake is as follows: 5–10 kg: 2–4 US cups (~500–1000 mL); 10–20 kg: 4–6 cups (~1000–1500 mL); 20–30 kg: 6–7 cups (~1500–1750 mL); 30–40 kg: 7–8 cups (~1750–2000 mL); 40–50 kg: 8–9 cups (~2000–2250 mL); >50 kg: 9–10 US cups (~2250–2500 mL) of water per day [104].

##### Physical Activity

Similar to diet, there is no unanimous opinion about the effects of increased physical activity on managing constipation, as there are a few studies in favor of it [105,106,107] and a few against it [108,109,110]. Despite the absence of a strong evidence base for these conservative interventions, they have been found to be useful in patients with NBD. Regular activity can help reduce constipation by stimulating the bowel’s peristaltic motility. It is important to encourage the young person to establish and continue a daily exercise program, which may include tailored wheelchair activities such as push-ups and transfers if necessary. A physical therapist can help develop such an exercise program, which is unlikely to do any harm and will have other health benefits for the child, even if bowel effects cannot be guaranteed.

##### Scheduled Defecation

We support the aim of establishing a pattern of scheduled defecation and exhausting the conservative interventions of dietary and lifestyle modification before moving on to pharmacological interventions. In general, to benefit from the diurnal “body clock,” scheduled defecation should be attempted once per day at approximately the same time every day (or, if not possible, on alternate days).

##### Maximizing the Gastrocolic Reflex

Another point to consider while setting the regimen is that the bowel contractions are maximal on waking up and after a meal or warm drink (the gastrocolic reflex). Although there is no strong evidence for its use in NBD [111,112], patients are still advised to make use of gastrocolic reflex by attempting to empty their bowels 10–30 min after eating or drinking [113]. For maximum effect, this can be combined with the scheduled defecation mentioned above.

##### Positioning

Several physical positions can encourage the passage of a bowel movement: placing the knees higher than hips, or the knees and hips bent in a typical squatting position. While no scientific studies exist on the use of specific commercially designed stools (e.g., Squatty Potty ^®^, LLC, St. George, UT, USA) the authors emphasize maximizing the squat position and adaptive seating to promote defecation. Additionally, in order to relax the pelvic floor muscles when sitting on the toilet, the feet should always be comfortably supported on a foot-stool, a customized orthopedic support, or the floor.

##### Abdominal Massage

Abdominal massage was used as a treatment for chronic constipation in the late 1800 s when there was a belief that it stimulated peristalsis [114]. Over the intervening years it fell out of favor but, with growing evidence in both children and adults, it has started regaining its popularity and it has reportedly been used beneficially by 22–30% of patients with NBD [4,115]. In a study of 24 adult patients with SCI, adding abdominal massage to the standard bowel program led to a significant reduction in colonic transit time (90.60 ± 32.67 h versus 72 ± 34.10 h, *p* = 0.035), abdominal distension (45.8% versus 12.5%, *p* = 0.008), and fecal incontinence (41.7% versus 16.7%, *p* = 0.031), while increasing the frequency of defecation (4.61 ± 2.17 versus 3.79 ± 2.15, *p* = 0.006) [116].

Despite the evidence showing this to be an effective technique, its mechanism of action is not entirely clear. Several observations have been noted and theories proposed, including activation of intestinal stretch receptors, which causes an increase in intestinal and rectal contraction [117], elicitation of measurable waves of rectal muscle contraction [118], decrease in colonic transit time [116], stimulation of the parasympathetic nervous system, thereby leading to an increase in gut secretions and motility and relaxing sphincters in the digestive tract [119]. In thin individuals, there may also be a direct mechanical effect. Whatever the mechanism, abdominal massage has a clear advantage of being non-invasive and risk-free, which is especially attractive to children, as well as repeatable and inexpensive. Abdominal massage in children is typically performed starting in the right iliac fossa, using a gentle, compressive, kneading motion in an upside-down “U” direction around the top of the umbilicus to the left iliac fossa, and then deep into the suprapubic region in order to help to move gas and stool along the course of the colon towards the rectum [114,120,121].

##### Digital Anorectal Stimulation

Digital anal/rectal stimulation [109,111,112,113] is a well-established technique used in individuals with NBD to help facilitate bowel evacuation. It requires the patient or caregiver to insert a gloved, lubricated finger into the rectum and move it in a rotatory pattern. This works by dilating the anal canal and relaxing the puborectalis muscle, which leads to a reduction in the anorectal angle. Both these effects lead to a reduction in resistance to the passage of stool, thereby assisting bowel emptying. Shafik et al. [122] in their study on 11 patients, noted left colonic contractions upon rectal distension which were absent after anesthetizing the rectum and anal canal. They therefore named this the rectocolic reflex [122], and it has been found to be useful in initiating bowel movement in individuals with supraconal disorders, but not in those with infraconal lesions. Overall, digital anorectal stimulation is a safe and effective intervention, with the main precaution advised to be gentle to avoid rectal mucosa injuries [122], especially if fingernails are long. This technique of digital stimulation is quite distinct from digital/manual evacuation, where the stool is extracted directly by the finger and which is generally not appropriate as a regular treatment for an older child.

##### Biofeedback and Physiotherapy

Biofeedback has anecdotally become quite popular in the treatment of various forms of fecal and urinary incontinence, including in children. In the early eighties, small case studies suggested a long-lasting beneficial effect of biofeedback in children aged 5–17 years with fecal incontinence secondary to myelomeningocele; in more than 50% of these children, fecal incontinence disappeared without the need for enemas or suppositories [123,124]. Larger controlled trials, however, showed insufficient effect of biofeedback alone in children with fecal incontinence due to spina bifida; patients allocated to behavior modification alone (defecation immediately after the evening meal each day, receiving a reward for defecating in the toilet without an enema or suppository, and undergoing an enema if unsuccessful for two consecutive days) showed a similar clinical improvement to patients allocated to behavior modification plus biofeedback [125,126] suggesting that the previous uncontrolled studies had overestimated the value of biofeedback in this population. Furthermore, biofeedback did not improve anal squeeze pressures or rectal sensation in these children [126]. Currently, therefore, it appears that there is no long-term advantage in adding biofeedback training to the conventional treatment of NBD in children, although there is limited evidence showing a short-term benefit in functional constipation [127].

Similarly, while (non-biofeedback) pelvic floor physiotherapy may be useful in some cases of functional constipation or bladder problems, there appears to be no literature justifying its routine use in children with NBD.

##### Non-invasive Electrical Stimulation

Normal bowel function depends on the passage of electrical impulses in sensory neurons from the rectum to the higher centers and returning via motor neurons to the anorectal muscles. Since this natural two-way communication process is interrupted in neuropathic conditions, investigators have attempted to restore the electrical milieu by providing replacement artificial electrical signals. Initial experimental treatment of neurogenic bladder/sphincter in adults and then children has been followed more recently by application of similar techniques to various bowel pathologies. Clearly, non-invasive techniques are preferable to surgical approaches, especially in children, but may not deliver the same potential therapeutic benefit.

##### Transcutaneous Electrical Nerve Stimulation

Non-invasive nerve stimulation such as transcutaneous electrical nerve stimulation (TENS) is widely used for bowel dysfunction in children: Veiga at al. in 2013 showed a 85.7% improvement in constipation in patients treated with para-sacral TENS [128]. TENS is well-accepted, safe, and studies suggest significant improvement in bowel function [129,130], although few have included a placebo-controlled group (essential for symptoms that are very open to psychological modification). Unfortunately, no specific data are presently available for NBD.

##### Posterior Tibial Nerve Stimulation

Posterior tibial nerve stimulation (PTNS) has reportedly improved the bowel dysfunction score in SCI [131], but no data or analysis have been presented specifically for neurogenic patients [132]

##### Other Electrostimulation

Other modalities of electrical stimulation have been suggested for NBD: transrectal and intravesical [133,134]. The limited experience reported does not yet permit these to be considered in daily clinical practice.

#### 3.2.4. Pharmacological Treatments

##### Probiotics

There is no specific evidence for the use of probiotics in children with NBD. However, the use of a probiotic can be considered for a general improvement in gut health and microbiome biodiversity. Probiotics can result in increased bowel frequency and improved bowel consistency in adults. When the probiotic *Lactobacillus reuteri* was administered to infants older than six months, there was improved bowel frequency [135].

##### Oral Laxatives

Oral laxatives are the next step up the ladder in the management of NBD. High-quality data exist in the form of several RCTs confirming the beneficial effect of laxatives in individuals with NBD. Polyethylene glycol (PEG)/macrogol has been found to be superior to lactulose in one RCT involving pediatric NBD [136], leading to higher bowel frequency (*p* < 0.01). Other commonly used oral laxatives include bisacodyl and senna (colonic stimulants), docusate (stool softener), and ispaghula husk (Fybogel, bulk-forming). While osmotic and stimulant laxatives form the mainstay of treatment in pediatrics, several different categories of oral laxatives are used. Lubricants such as mineral oil can also be used to help with passage of hard stool.

Osmotic laxatives used to improve the consistency of hard stool (types 1 or 2 on the Bristol Stool Chart [137]:

Lactulose (10 g/15 mL suspension)—response may take 24–48 h.

Polyethylene glycol (PEG)/macrogol 3350 (powder mixed with water) 0.2–1.5 g/kg/day; onset of action is 24–96 h.

There is no recommendation from the literature on the optimum number of doses per day of lactulose and PEG. However, compliance with any laxative improves if prescribed only once per day [138]. On the other hand, PEG involves a large volume of liquid to be ingested (typically 125 mL per adult sachet, or half of this per pediatric sachet), so many doctors dose at least twice per day.

Milk of Magnesia:2–5 years old: 0.4–1.2 g/day, in 1 or more doses;6–11 years old: 1.2–2.4 g/day, in 1 or more doses;12–18 years old: 2.4–4.8 g/day, in 1 or more doses.

Stimulant Laxatives Used to Increase Frequency of Bowel Movements Through Intestinal Contraction:

Bisacodyl:2–10 years old: 5 mg once per day;10–18 years old: 5–10 mg once per day.

Sennosides (Docusate Sodium oral suspension or Sennosides tablets)—onset of action is 6–10 h:2–6 years old: 2.5–5 mg/day in 1–2 doses;6–12 years old: 7.5–10 mg/day in 1–2 doses;12–18 years old: 15–20 mg/day in 1–2 doses.

Initial laxative doses suggested to families are intended only as a guide: the response of pediatric neurogenic bowel to laxatives can be quite variable, so parents or caregivers need to be advised to titrate up or down the starting doses of osmotic and stimulant laxatives separately every few days until their child achieves just the right stool consistency and frequency respectively. Similarly variable can be the timing between ingestion and action of an oral laxative in children with NBD so that, especially for those with poor anal sphincter control, it can be difficult to ensure that the resulting evacuation does not occur at a socially inconvenient time such as during school. This tends to limit the usefulness of oral laxatives in children with NBD.

##### Suppositories

If digital stimulation is not effective in providing the desired symptomatic relief, or rectal lesion occurred [139] it can be augmented by the use of laxative suppositories glycerin(e)/glycerol and bisacodyl are the commonly used suppositories, with the former mild enough to be used in infants, but often ineffective in older children with NBD. The latter is a stimulant laxative that has either hydrogenated vegetable oil or polyethylene glycol (PEG) as a base; three studies (including one good-quality randomized controlled trial, RCT) have reported better results with PEG-based suppositories [140,141,142]. Sodium bicarbonate (Lecicarbon) is a newer effervescent suppository, releasing bubbles of carbon dioxide to stimulate reflex rectal activity, which has a quicker onset of action than fat-based bisacodyl suppositories (15–20 min compared to 30–40 min) but similar efficacy [143].

In children with a patulous anus (often seen in spina bifida), the suppository sometimes falls out before it has had a chance to work. This can often be resolved by holding the buttocks together and/or encouraging the child to lie prone and stay relatively still until they feel contractions or start to stool.

##### Enemas

An enema is an instillation of liquid into the rectum to evacuate stool. Although enemas are often used for acute constipation in people with neurotypical bowels, regular enemas can form part of an effective bowel management program for people with NBD. They are generally used in the event of suppositories being unproductive.

The two main approaches are to either deliver a relatively large volume of water or saline into the colon to produce a mechanical flush, or to use commercially available micro-enema tubes, usually containing 5 mL of a strong stimulant laxative to act locally. The former approach will be considered below as transanal irrigation. In the latter category, docusate sodium mini-enema has been shown to be more effective in NBD than glycerine or bisacodyl suppositories [140]. The other commonly used micro-enemas are sodium citrate and sorbitol. Sodium phosphate enemas, on the other hand, contain a medium volume of laxative (60 mL for ages 5–11 years, and 120 mL for ages 12+), so can be convenient and effective in occasional acute cases of fecal impaction in an otherwise well child. However, phosphate enemas are not routinely used on a regular basis as part of a bowel program, as they can be messy, and risk dehydration and electrolyte abnormalities secondary to inadvertent retention of the medication if they do not produce a stool within 10 min (a particular risk in children with megacolon or with renal compromise due to an associated neurogenic bladder); long-term use can also cause colitis due to chronic irritation, sometimes leading to diarrhea secondary to a narrow hyperactive colon [142]. However, this complication has recently been reported in both children and adults using other antegrade and retrograde enema irrigants too [143].

The tip of any enema tube can injure the fragile rectal mucosa, especially in a child. As with suppositories, the liquid medication in an enema can also sometimes dribble out prematurely past a lax anus, reducing its efficacy.

##### Transanal Irrigation

When a micro-enema tube is not being used, there are various options for delivery of a larger volume of liquid into the rectum and colon:Bulb syringe enemas are used for smaller volume enemas in older infants and young toddlers. The bulb is inserted through the anus and 60–90 mL of warm water can be instilled.Balloon enemas use a 24 Fr Foley catheter to administer a high-volume rectal enema. To have the enema administered, the patient must usually lie down, with the catheter’s balloon inflated inside the lower rectum to create a leak-proof seal above the anal canal. The individual must then transfer to the toilet to deflate the balloon and evacuate at the appropriate time, all of which can be quite challenging for a child who may have other disabilities.Cone enemas involve insertion of the tip of a graduated silicone cone until it occludes the anus (whether patulous or not) with a water-tight seal. This is simpler, less cumbersome, and somewhat less invasive than a balloon catheter and so may better suit younger children. The cone is connected to enema tubing in a similar manner to balloon enemas. Afterwards, the cone and tubing can easily be washed and re-used, making it relatively inexpensive compared to balloon enemas and specifically designed kits.Commercially available transanal irrigation (TAI) systems were designed to speed up the colonic washout process and increase independence in bowel management compared to the previous generic balloon catheter and cone techniques. All incorporate either a customized bag or chamber from where the irrigant solution drains along a tube ending in either a catheter or a cone that is passed through the anus to administer high-volume enemas over several minutes that have been shown on scintigraphy to clear far enough up the colon to render someone reliably clean for a few days [144].

In balloon catheter TAI systems, a rectal catheter is intended to stay in place without assistance while the enema fluid is released via a pump operated either manually (e.g., Peristeen by Coloplast^®^ A/S (Kokkedal, Denmark) or IrriSedo Klick by Qufora^®^ (Allerod, Denmark) or electrically (e.g., Navina by Wellspect^®^, Molndal, Sweden). However, those patients with a lax anal sphincter generally find the washout is interrupted prematurely when the balloon is inadvertently expelled intact (whether by gravity, recoil, or rectal contraction). Therefore, they depend on the catheter being held in place manually. Younger children, and those whose neurological condition also affects their balance or manual dexterity, are generally unable to achieve this for themselves, so require a caregiver to be present throughout the few minutes when the fluid is being administered.

In cone TAI systems, the silicone cone tip, where the irrigation fluid meets the body, must be held inside the anus throughout the delivery of the irrigant in order to plug the fluid from being expelled prematurely. In patients with a patulous anus, the graduated shape of the cone tends to achieve a more effective seal than a balloon catheter, but again a carer is often required to hold the cone in place. The various cone systems use gravity (e.g., Assura by Coloplast^®^), a manual pump (e.g., IrriSedo Cone by Qufora^®^ or Peristeen Cone by Coloplast^®^), or an electric pump (e.g., IryPump by B Braun^®^ Melsungen, Germany) to drive the irrigation.

Either way, the washout can be performed completely on the toilet (or on a commode or shower-chair in a suitable wet-room) or may involve transferring from the floor/bench to the toilet before its onset of action a few minutes later. Complete emptying of all the irrigant and the accompanying stool takes up to one hour on the toilet, depending on the volume of fluid used, which determines how far proximally the colon is cleared, and so how many days the child can remain clean for afterwards. This emptying process can be augmented by abdominal massage (see Section Abdominal Massage above).

This method of irrigation has been used clinically starting in 1987 to treat constipation and fecal incontinence in children [145]. In children with NBD who do not respond to conservative or medical treatments, TAI is an increasingly accepted treatment [13]. There are studies demonstrating improved QoL and outcomes with children utilizing TAI for functional incontinence and functional constipation, and some authors recommend TAI should be mandated prior to considering any invasive surgical intervention [146]. A summary publication of TAI use in children incorporated a literature review comprising 27 studies with 1040 patients whose average age was 8 years old [13]. Of these children, 78–84% had improved bowel continence, and 95% had improved QoL, after starting TAI. There are undoubtedly some children who, even as adults, will never achieve independence with TAI, particularly due to their body shape and neurological function. However, in some units TAI has largely replaced the traditional surgical approach for individuals who fail to respond to escalating conservative and medical treatments, as it has proven to be as effective as surgery without the additional morbidity.

Whatever the mode of delivery, the irrigant solutions used will vary based on individual needs. The vast majority of prescribing clinicians in Europe and North America recommend tap water as the irrigant of choice. However, this introduces the possibility of the hypotonic water being absorbed by the colon, producing a theoretical risk of iatrogenic hyponatremia. The number of published reports of successful and safe colonic irrigation including TAI using simple tap water suggest that such concerns are probably unfounded. Nonetheless, to counter this risk, some units instead irrigate with normal (0.9%) saline, either commercially prepared or approximated at home by adding 9 g (1.5 teaspoons) of standard table salt (but not low-salt/low-sodium preparations) to each one liter of tap-water. However, this approach carries its own risk of errors in parental understanding and titration. Water or saline alone may be sufficient for an enema program but, if not successful, relatively gentle additives can make the enema more effective. Examples include baby soap (contains glycerine as an ingredient), USP-grade glycerin (used as a stimulant laxative), Castile soap (considered “stronger” than glycerin or baby soap), or PEG in the enema fluid (instead of, or in addition to, taking PEG orally). For those patients preferring a longer interval between washouts, and willing to accept a longer sit on the toilet or the possibility of increased abdominal cramps, a larger volume of irrigant (up to 20 mL/kg body weight) can be instilled and/or a stronger stimulant laxative such as bisacodyl can be added to the liquid. Conversely, for other children and families their priority is as short a TAI session as possible, so they use a smaller volume of liquid every day.

##### Summary

Many therapeutic approaches exist for the management of NBD in children and adolescents. Treatment must be tailored to the needs and circumstances of the individual child and their caregivers. Although isolated strategies may act as a starting point, more often than not, the management becomes multidimensional, involving different treatment modalities. To be maximally effective, a bowel management program in pediatric NBD must also be multidisciplinary, involving close and long-term teamwork between the child with their parents or caregivers and a wide range of specialists including continence nurse specialists/uro-therapists, school and community nurses, family doctor, physical therapist, pediatricians, pediatric gastroenterologists, pediatric clinical psychologists, radiologists, pediatric surgeons/urologists, and pediatric anesthesiologists.

### 3.3. The Surgical Management of NBD in Children and Adolescents

#### 3.3.1. Sacral Nerve Modulation

Sacral nerve modulation (SNM) is a step up from transcutaneous electrostimulation techniques, involving invasive implantation of electrodes along sacral nerve roots, which brings more targeted effects (i.e., it is possible to focus on either the rectum or the anal sphincter or both) but this is balanced by higher risks of nerve damage and introducing infection. SNM was initially developed to control lower urinary tract symptoms, primarily in neuropathic conditions, and has more recently been used for bowel dysfunction too. SNM works by stimulating the somatic and autonomic nervous systems, although the exact mode of action is not completely understood [147,148] and few studies have proposed its effect on the central nervous system [138]. Its impact in cases of constipation has been suggested to be due to an increased frequency and amplitude of antegrade pressure sequences, but whether these are mediated via a central or peripheral mode of action remains unclear. In adults, randomized controlled trials of SNM in chronic constipation have not shown benefit [149], so it is currently indicated only for fecal incontinence. In children and young adults with refractory functional constipation, SNM has shown some sustained benefit, although it is debatable whether this is enough to justify the risks and high costs [150]. However, SNM is not FDA-approved in the USA for bowel dysfunction in children under the age of 18 (nor under the age of 16 for bladder dysfunction). Furthermore, it may not be technically feasible for the more common causes of NBD which involve anatomical abnormality of the spinal cord, such as spina bifida and spinal cord injury. The role of SNM in neurogenic patients has been evaluated in a few studies and improvement has been reported in SCI [151].

#### 3.3.2. Bowel Surgery

Surgical management of NBD is regarded as a valuable option in selected cases [152]. With respect to the treatment pyramid of pediatric NBD, chronic constipation and/or fecal incontinence that was previously proposed by our group (see modification in Figure 1), surgical methods for bowel management are usually utilized only after failure of the full range of conventional conservative and pharmacological medical treatments, which now includes transanal irrigation (TAI) [13]. Nevertheless, a recent survey showed that, in order to achieve fecal control, surgery is required (due to failure of medical treatment) in about 40% of pediatric and adult patients with NBD secondary to myelomeningocele [153]. Several studies show that surgical treatment of NBD can be successful in providing an improved QoL if appropriately indicated and with patients carefully selected. The aim of surgery for NBD, just as with its conservative and pharmacological management, is to evacuate the colon at a time and place of the child and family’s choosing, in order to reduce the prospect of soiling at times when the child is unable to visit the toilet. It should also minimize the average time the child needs to spend in the bathroom every week.

So far, most reports on surgical treatment of NBD deal with adult patients and very little has been published on children and adolescents on this topic. However, most of the benefits and drawbacks of surgery in these patients apply to all age groups. Nevertheless, it is important to remember that young patients are still growing (physically and emotionally), and probably need to continue for life with the surgical established method for emptying their colon. The proposed surgical options also have to respect the pediatric patient’s developmental age and any comorbidities, as well as the family dynamics and environment, in order to produce an appropriate individualized solution that allows optimum social integration with their age-appropriate peers [154].

The surgical approach for NBD in children primarily involves creating artificial “upstream” access for antegrade administration of colonic irrigation enemas, either by Malone’s antegrade continence enema (ACE) procedure, or by tube cecostomy. This might be especially advantageous in patients with stool impaction due to NBD [155] or in those who, due to comorbidities, lack the balance, manual dexterity, or motivation to self-administer retrograde washouts by TAI [13]. Many teenagers can administer their antegrade enemas independently via an intermittently inserted catheter or an indwelling tube. The final surgical alternative in children is a colostomy (fecal diversion), but Malone´s ACE procedure is by far the most utilized method [156]. Unfortunately, some other reconstructive techniques available to adults with NBD, such as artificial anal sphincter implantation [157], are generally not appropriate for the growing child.

##### Malone Antegrade Continence Enema Procedure

Malone´s ACE procedure has been shown to be a safe surgical method, with minimal mortality but several minor complications [158]. The successful use of the Malone antegrade continence enema (MACE) via a neo-appendicostomy has increased QoL in 80% of adult patients [159]. The MACE has also been successfully implemented in children with spina bifida and resulted in a significant improvement in fecal continence and QoL scores [160,161].

The current standard in situ appendicostomy for the MACE produces a continent catheterizable appendiceal channel to the cecum by creating a valve mechanism at the cecal end (to reduce leakage of feces onto the skin) and bringing the decapitated end of the appendix up to a convenient site on the abdominal wall such as the umbilicus or hidden under a cosmetic skin-flap elsewhere that also serves to reduce the risk of stomal stenosis. Beside this technique, other open surgical modifications have been performed in the pediatric age group such as the cecal extension (when the appendix is not long enough), the Yang-Monti ileo-cecostomy (using a short section of detubularized retubularized ileum to create an alternative channel when a suitable appendix is not available) and cecal or colon flap channels (again if an adequate appendix is not available) [161]. MACE channels are often constructed at the same time as urinary reconstructive surgery such as a Mitrofanoff procedure for associated neurogenic bladder. If the appendix is not long enough, or cannot be extended sufficiently, to create both channels, this may give rise to surgical dilemmas regarding the optimum use of the appendix, and the need to use such modifications. However, the rate of surgical revisions required after some of these modifications appears to be higher than for a standard MACE [162]. In the subsequent laparoscopic adaptation, there is usually no attempt at the technically difficult creation of a valve mechanism, yet the rates of fecal leakage via such stomas are still surprisingly low [163,164].

If investigation such as a colonic transit study suggests mega-rectum and/or distal colonic delay with feces impacting in the recto-sigmoid, then a “distal ACE” (e.g., in the transverse or descending colon) can produce a more effective evacuation of feces and reduce the risk of retention of the irrigant compared to the conventional cecal ACE [165].

##### Tube Cecostomy

Another modification of the MACE is the utilization of a Chait^®^ (Cook Medical LLC, IN, USA) cecostomy tube, or a “button” device, placed as either a percutaneous endoscopic cecostomy (PEC), under fluoroscopic guidance, or via laparoscopy. It has been proven significantly to improve fecal continence and QoL in patients with NBD [166]. The disadvantage is that any such tube needs to be replaced at regular intervals, and sooner if it blocks, dislodges, or breaks.

As with a conventional ACE, in cases of slow colonic transit the Chait^®^ tube or button device can instead be placed more distally in the colon (e.g., at the descending/sigmoid junction) as a percutaneous endoscopic colostomy [167].

Outcomes of MACE and tube cecostomy are comparable in children with spina bifida (SB) [154]. Nonetheless, both procedures, however performed, carry the important potential risk of jeopardizing the critical ventriculo-peritoneal shunt in children with hydrocephalus associated with spina bifida [168].

##### Bowel Diversion

A colostomy involves bringing part of the large intestine to the abdomen’s surface to form a stoma. Stool is collected in an external bag worn by the patient over the stoma. Perhaps the main barrier to performing a bowel stoma in any age group is the reluctance of the patient to accept it from a psychological perspective. This is particularly relevant in the pediatric population, where children and parents may fear leakage of feces, flatus, or smell, its impact on bodily integrity and self-image, and the possibility of teasing or bullying by peers. However, ostomy (either colostomy or ileostomy) as a bowel diversion produces similar or even superior outcomes in selected patients in regard to QoL compared to conservative bowel management strategies in NBD. For those individuals who prefer their stoma to act at a convenient time, the upstream colon can be irrigated retrogradely in a similar fashion to TAI (see Section Transanal Irrigation.). Nevertheless, a relevant number of postoperative complications has been reported. The main advantage of diversion is the reduction of time taken to empty the bowel. Patients who undergo ostomy surgery, often as a “last resort,” are usually very satisfied with the resulting improvement, and a significant proportion of patients afterwards report a desire to have been counselled about this option earlier [169] rather than reserving it for supposed failure of care [170,171]. Furthermore, adult series of colostomy in fecal incontinence showed a reduced number of hospitalizations [172]. Colostomy formation early after spinal cord injury has also been shown to improve independence and increase acceptability of bowel management [173,174]. Occasionally ostomy is mandated in order to divert the fecal stream from the perineum so that chronic decubitus pressure-sores may heal without being soiled.

##### Bowel Resection

Bowel resection has been proposed for selected cases of functional constipation and/or fecal incontinence after failed conservative treatment [175,176]. Outcomes in these children were reported to be favorable in most (up to 80%) of the cases [177]. However, bowel resection does not play a role in the surgical treatment of NBD apart from occasional limited resections during creation of MACE or ostomy [178]. Some authorities recommend routine consideration of bowel resection at the time of MACE creation to encourage faster and more complete bowel evacuation. On the other hand, others suggest that bowel resection should be reserved for the few individual cases where there is a strong indication. However, while this is controversial topic, currently there is no common consensus to resect bowel at the time of MACE creation.

##### Summary

For children born with NBD, fecal continence can be achieved in about 50% with conservative and medical treatment, although the advent of more user-friendly versions of TAI promises to raise this proportion. The vast majority of the remaining patients should also reach fecal continence by undergoing one of several possible surgical interventions, most often the MACE. Nevertheless, all surgical procedures carry a risk of postoperative complications and revisionary surgery, which can be especially difficult to deal with for children and adolescents and their families. Therefore, surgical treatment in pediatric NBD should be offered only on an individually indicated basis [158].

## 4. Discussion

Today the majority of pediatric patients with NBD can theoretically achieve social fecal continence and treat chronic constipation, reducing their need for pads and time spent in the bathroom, enriching their QoL and social relationships, improving their productivity in school/work, and reducing the incidence of related urinary tract infections. In the past, this goal has often only been achieved by resorting to surgical procedures such as the Malone ACE, or even ostomy. Such traditional surgical continence procedures can indeed be highly effective in carefully selected patient groups [155], but they carry a relatively high risk of surgical complications and an increased risk of anesthesiology procedures. The advent of TAI changed traditional bowel management for the significant numbers who do not respond to conservative and pharmacological approaches alone, permitting successful treatment of a large population of pediatric and adolescent patients with NBD, without requiring surgery. TAI can be frustratingly time-consuming for impatient children, but has been shown to reduce total time spent in the bathroom dealing with the effects of constipation and/or incontinence. Of course, TAI must be tailored to different patient populations and individual requirements in order to obtain a good outcome for the child and their caregivers [13]. Indeed, TAI may not be feasible for certain individuals with reduced hand control, poor balance, or distorted spines who wish to be independent in their bowel evacuation. In those circumstances, surgery instead can be life-enhancing. However, TAI now forms part of a thorough bowel management program that must first include a range of conventional conservative measures (such as physical activity, correct fluid intake and diet, probiotics, etc.), followed by the addition of laxative medication administered orally and/or per rectum. All the therapeutic strategies for bowel management must be individually tailored to each different child and family, considering the pathophysiology of their neurogenic bowel dysfunction, their primary disease and associated comorbidities, emotional, educational, or mental status, manual dexterity, as well as the fears, motivation, and compliance of the child and caregivers. Today invasive surgical treatment is usually postponed and used only in case of failure of conservative, pharmacological and mini-invasive (i.e., TAI) treatment applied in a stepwise approach as recommended by our group [13] and by the International Children’s Continence Society [18]. For this reason, the indignity of NBD must be addressed in all pediatric populations with neurological conditions, including patients with severe disabilities such as cerebral palsy, acquired brain injury for trauma, tumor, vascular injury or systemic disease that, until now, have not been afforded the same therapeutic attention as spina bifida or traumatic spinal injury.

## 5. Conclusions

NBD today should be considered, investigated, and treated in all children and adolescents with any congenital or acquired neurological disease, with a high expectation of success. Bowel management should be tailored to the individual needs and circumstances of each patient and their family, and all conventional conservative and medical treatments must be exhausted before considering proceeding to a surgical approach. A structured but aggressive approach to the treatment of NBD should improve distressing symptoms, enhance QoL for the child and their caregivers, and will decrease hospital readmissions. TAI seems a very effective bridge between conservative/medical management, that is only effective in about half of affected children, and definitive surgery that is much more effective for most but carries unwanted risks.

## Figures and Tables

**Figure 1 jcm-10-01669-f001:**
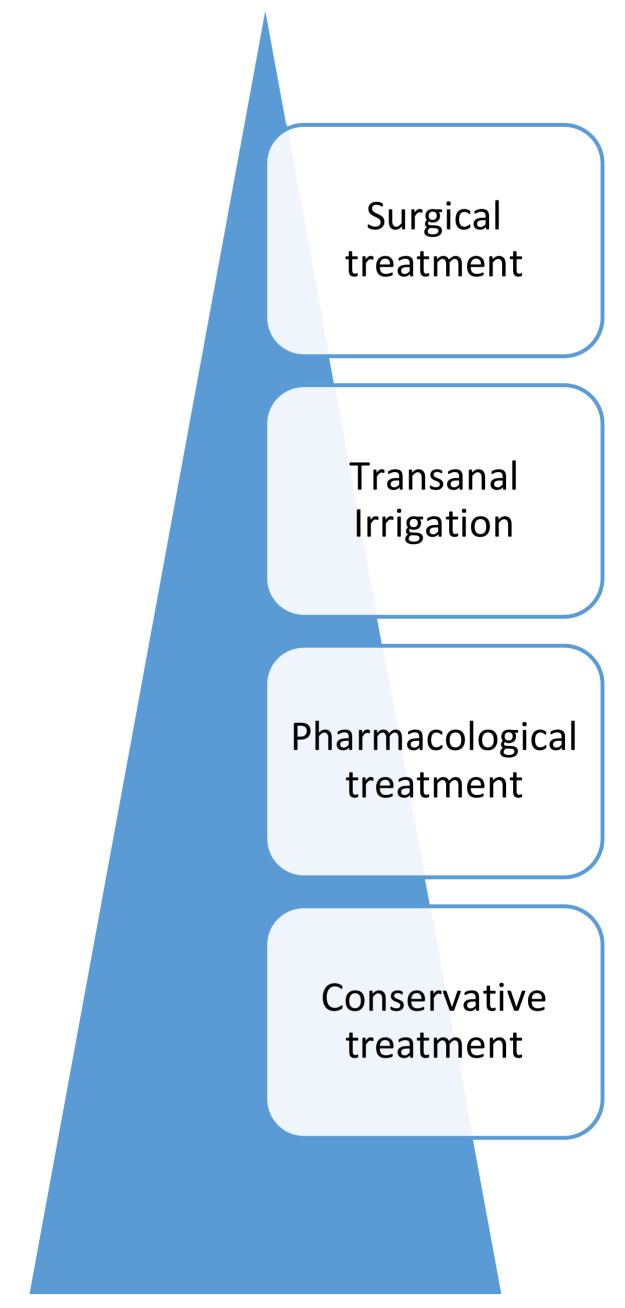
Pyramid of treatment recommendations for neurogenic bowel dysfunction, adapted from Mosiello et al. [13].

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
