# Peer review of "Neurogenic Bowel Dysfunction in Children and Adolescents"

_jcm, 2021, doi:10.3390/jcm10081669_

Round 1

Reviewer 1 Report

This is a very thorough paper on management of neurogenic bowel dysfunction (NBD) and is really one of the best documents that I have read on the topic.  The authors are to be commended on the result of this collaborative multi-disciplinary project.

I am bothered by the authors’ use of the term “consensus. “ This makes it sound as if the proposed management of NBD was conceived and endorsed by medical societies.  However, the authors were brought together on an advisory board by Coloplast that manufactures a widely-used TAI product and paid for publication costs.  There is a disclosure but it is only reached after reading to 22 pages.  Many journals place the disclosure at the beginning of articles and that seems more appropriate in this case, particularly because there is a rather strong endorsement of TAI usage in the manuscript.

Specific comments:

Page 2, line 49-50.  Treating constipation is more beneficial to the urinary tract than just improving functional bladder capacity (by reducing rectal compression of the bladder) and reducing UTI by fecal soilage. Treating constipation improves bladder function (proven urodynamically) because rectal distension affects pelvic nerve function due to cross-talk between the rectum and bladder.  Mechanic compression may not be that critical; the uterus lies between the rectum and bladder in females. Treating constipation not only improves bladder capacity but reduces detrusor overactivity so that urinary continence is improved. Treatment of constipation in the absence of encopresis reduces UTIs by improving bladder function (not merely by reducing soiling of the perineum).

Methods should include more disclosure that the authors are consultants to Coloplast and that Coloplast underwrote publication costs.

Page 3 Line 122 – It is more than just ingestion of folic acid (which would seem to be an individual choice) that has reduced incidence of spina bifida but widespread folic acid dietary supplementation.

Page 3, Line 125 – It may clearer to the reader to explain that 93% of MMC (from line 107) affects the lumbosacral spinal cord that innervates the bladder, lower colon, and sphincters which is why neurogenic bladder and bowel dysfunction are nearly universal in this population.

Page 4, line 167 – It may clearer to the reader to explain that the ultrasound scan of the spinal cord is performed to look for a low-lying conus below L2.

Page 4, line 171 – It is not clear if the authors are implying the congenital defect of ARMs is the intestinal malformation or the congenital neuropathy (or both) as a contributing factor to NBD.

Page 4 – It may helpful to the reader to clarify that cerebral palsy does not involve a bowel or sphincter neuropathy but higher level control of these structures.

Page 4 – It may be helpful to the reader to give more information if bowel dysfunction in muscular and mitochrondrial disorders is due to a neuropathy, myopathy, or developmental delay.

Page 5, line 227-231.  It seems a bit out of place to discuss obstetical trauma as a cause of NBD in a manuscript on children and adolescents.

Page 5, line 243-4.  It would be helpful to the reader to know if degree of constipation correlates more with higher or lower level SCI.

The sections of cauda equina syndrome and spinal canal stenosis would seem to be more logical following acquired SCI because of their similarities in outcomes and in not being congenital.

Page 7, line 303.  Does the referenced study apply to children or adults?

Page 8, line 370  It may be helpful to let the reader know that NBD is not only missed but often ignored.

Section 3.1.2.1 Impact of anatomic location.  Urologists stress that malformation or injury to the sacral 2-4 nerve roots results in sphincteric dysfunction and often flaccid sphincter, leading to urinary incontinence.  This physiology is not specifically covered for NBD in this manuscript.

Section 3.2.3 Conservative and medical treatments – these two forms of treatment should be separated.

The discussion of fiber on page 13 is excellent.  It may helpful to give age or weight-based suggestions of fiber intake as is done for the medications.  The differentiation of water-soluble and insoluble fiber is intriguing, but the reader would be helped by specific examples of each type.

Page 14 – Positioning – is there any data on stools/chair specifically designed for defecation – i.e, Squatty Potty?

Page 14, line 644-5 Was the study on abdominal message in children or adults?

Page 15 – Biofeedback – is there data on biofeedback in NBD or just in neuro-normal children?

Page 15, line 688-9 – is there a recommendation on number of doses per day of Lactulose and PEG as they are for other medications?

Page 17 – Cone enemas are presented but more as part of a commercially available system.  However, in younger children, simple cones attached to enema tubing are simpler and less invasive than balloon catheters (line 775) and deserve mention.  They work more like balloon catheters and are effective and inexpensive.

Page 17 line 824 – water or saline.  There is a great deal of controversy about using tap water versus saline for high volume enemas.  Some discussion of this should be mentioned and perhaps as well as a recipe for making inexpensive saline solutions at home.

The discussion of sacral nerve stimulation on page 19 should be more specific that SMS is FDA not approved under age 16 for bladder dysfunction and age 18 for bowel dysfunction.  Also, it may not be appropriate for the more common causes of NBD such as SB and SCI.

Page 20, line 932.  Many MACEs are placed in the umbilicus.

In the discussion of MACE, it should be mentioned that many MACE channels are created at the time of surgery for neurogenic bladder when the appendix may also be needed for creation of a Mitrofanoff channel, creating surgical dilemmas on how to best use the appendix in such cases.

Page 20 – Tube cecostomy discussion should mention the disadvantages compared to MACE in that the tube has to be replaced regularly and can be dislodged.  Likewise, the discussion of risk to the VP shunt holds true for MACE procedures as well.

Page 20 – the discussion of colostomy should also include that colostomy are sometimes medically necessary for management of sacral decubiti to prevent wound soiling.

Page 21 – Bowel resection – Followers of Levitt often recommend bowel resection at the time of MACE creation to allow more complete and faster bowel evacuation.  Certainly, it is controversial topic.

Discussion – I felt that the discussion was very pro-TAI even before I read the conflicts.  TAI is good and has had a great effect on NBD management but has many limitations.  Reference 141 demonstrates that is not as effective for continence as antegrade enemas.

Author Response

Dear Editor, Dear Reviewers,

Here enclosed are our corrections according to Your valuable suggestions.

May thanks for Your time and improvements

On Behalf of all Authors

Comments and Suggestions for Authors

This is a very thorough paper on management of neurogenic bowel dysfunction (NBD) and is really one of the best documents that I have read on the topic.  The authors are to be commended on the result of this collaborative multi-disciplinary project. Thank You very much

I am bothered by the authors’ use of the term “consensus. “ This makes it sound as if the proposed management of NBD was conceived and endorsed by medical societies.  However, the authors were brought together on an advisory board by Coloplast that manufactures a widely-used TAI product and paid for publication costs.  There is a disclosure but it is only reached after reading to 22 pages.  Many journals place the disclosure at the beginning of articles and that seems more appropriate in this case, particularly because there is a rather strong endorsement of TAI usage in the manuscript.

 All mentions of consensus-group/consensus-statement removed. Raw 21-23, 63, 93

Additional disclosure statement added to start of Methods section,,

raw 71

Specific comments:

Page 2, line 49-50.  Treating constipation is more beneficial to the urinary tract than just improving functional bladder capacity (by reducing rectal compression of the bladder) and reducing UTI by fecal soilage. Treating constipation improves bladder function (proven urodynamically) because rectal distension affects pelvic nerve function due to cross-talk between the rectum and bladder.  Mechanic compression may not be that critical; the uterus lies between the rectum and bladder in females. Treating constipation not only improves bladder capacity but reduces detrusor overactivity so that urinary continence is improved. Treatment of constipation in the absence of encopresis reduces UTIs by improving bladder function (not merely by reducing soiling of the perineum).

Amended r 49-52

Methods should include more disclosure that the authors are consultants to Coloplast and that Coloplast underwrote publication costs.

Additional disclosure statement added to start of Methods section.raw 73…..

Page 3 Line 122 – It is more than just ingestion of folic acid (which would seem to be an individual choice) that has reduced incidence of spina bifida but widespread folic acid dietary supplementation.Amended 131-135

Page 3, Line 125 – It may clearer to the reader to explain that 93% of MMC (from line 107) affects the lumbosacral spinal cord that innervates the bladder, lower colon, and sphincters which is why neurogenic bladder and bowel dysfunction are nearly universal in this population.Amended, r136-137

Page 4, line 167 – It may clearer to the reader to explain that the ultrasound scan of the spinal cord is performed to look for a low-lying conus below L2. Previous sentence amended to include this.

Page 4, line 171 – It is not clear if the authors are implying the congenital defect of ARMs is the intestinal malformation or the congenital neuropathy (or both) as a contributing factor to NBD. Clarified, r 187

Page 4 – It may helpful to the reader to clarify that cerebral palsy does not involve a bowel or sphincter neuropathy but higher level control of these structures. Amended., r198-199

Page 4 – It may be helpful to the reader to give more information if bowel dysfunction in muscular and mitochrondrial disorders is due to a neuropathy, myopathy, or developmental delay. Amended, 222-230

Page 5, line 227-231.(264-265)  It seems a bit out of place to discuss obstetrical trauma as a cause of NBD in a manuscript on children and adolescents.Clarified

Page 5, line 243-4.  It would be helpful to the reader to know if degree of constipation correlates more with higher or lower level SCI. Clarified, now280-295

The sections of cauda equina syndrome and spinal canal stenosis would seem to be more logical following acquired SCI because of their similarities in outcomes and in not being congenital. Thanks for the suggestion , We discussed about that and We would like to maintain our original classification, defined in our experience for relevance for pediatric patients. Please consider that this is our opinion and not a real classification for frequency, where some causes are rare disease , sometimes not recognized or treated, and difficult to define in a correct order.

Page 7, line 303.  Does the referenced study apply to children or adults? Specified,r 358

Page 8, line 370  It may be helpful to let the reader know that NBD is not only missed but often ignored. Amended

Section 3.1.2.1 Impact of anatomic location.  Urologists stress that malformation or injury to the sacral 2-4 nerve roots results in sphincteric dysfunction and often flaccid sphincter, leading to urinary incontinence.  This physiology is not specifically covered for NBD in this manuscript. Amended

 And section 3.1.2.1 covers this more in detail- please let me know if I am missing something ra 280-295

Section 3.2.3 Conservative and medical treatments – these two forms of treatment should be separated. Done (with some re-ordering & re-numbering of sections).

The discussion of fiber on page 13 is excellent.  It may helpful to give age or weight-based suggestions of fiber intake as is done for the medications.  The differentiation of water-soluble and insoluble fiber is intriguing, but the reader would be helped by specific examples of each type.

Can add the following section to the paragraph:done

Soluble fiber includes plant pectins and gums commonly found in foods like lentils, peas, oats, barley, apples and citrus foods. Insoluble fiber includes plant cellulose and hemicellulose including whole wheat or bran products, green beans, potatoes, cauliflowers and nuts.

Recommended Fiber Dosage (g/day): 0-12 months: not determined, 1-3 years: 19 grams, 4-8 years: 25 grams, 9-13 years: 26 to 31 grams, 14 years and older: 29 to 38 grams

Page 14 – Positioning – is there any data on stools/chair specifically designed for defecation – i.e, Squatty Potty?

While no studies on the use of specific stools exist, the authors emphasize maximizing the squat position using commercially designed stools and adaptive seating to promote defecation.

Page 14, line 644-5 Was the study on abdominal message in children or adults?

Studies supporting abdominal massage found both in adult and children

Page 15 – Biofeedback – is there data on biofeedback in NBD or just in neuro-normal children?

  • Amended Use of biofeedback in treatment of fecal incontinence in patients with meningomyelocele.

Wald A.Pediatrics. 1981 Jul;68(1):45-9.PMID: 7243508

  • Treatment of fecal incontinence in children with spina bifida: comparison of biofeedback and behavior modification.

Whitehead WE, Parker L, Bosmajian L, Morrill-Corbin ED, Middaugh S, Garwood M, Cataldo MF, Freeman J.Arch Phys Med Rehabil. 1986 Apr;67(4):218-24.PMID: 3964054

  • Biofeedback for neurogenic fecal incontinence: rectal sensation is a determinant of outcome.

Wald A.J Pediatr Gastroenterol Nutr. 1983 May;2(2):302-6.PMID: 6875754

  • [Biofeedback in faecal incontinence].

Kroesen AJ, Buhr HJ.Chirurg. 2003 Jan;74(1):33-41. doi: 10.1007/s00104-002-0567-5.PMID: 12552403 Review. German.

  • Biofeedback therapy for anorectal disorders in children.

Berquist WE.Semin Pediatr Surg. 1995 Feb;4(1):48-53.PMID: 7728508 Review.

Page 15, line 688-9 – is there a recommendation on number of doses per day of Lactulose and PEG as they are for other medications?

Adherence to Polyethylene Glycol Treatment in Children with Functional Constipation Is Associated with Parental Illness Perceptions, Satisfaction with Treatment, and Perceived Treatment Convenience.

Koppen IJN, van Wassenaer EA, Barendsen RW, Brand PL, Benninga MA.J Pediatr. 2018 Aug;199:132-139

Page 17 – Cone enemas are presented but more as part of a commercially available system.  However, in younger children, simple cones attached to enema tubing are simpler and less invasive than balloon catheters (line 775) and deserve mention.  They work more like balloon catheters and are effective and inexpensive.Added a new part r924-929

Page 17 line 824 – water or saline.  There is a great deal of controversy about using tap water versus saline for high volume enemas.  Some discussion of this should be mentioned and perhaps as well as a recipe for making inexpensive saline solutions at home.added r 970-987

The discussion of sacral nerve stimulation on page 19 should be more specific that SMS is FDA not approved under age 16 for bladder dysfunction and age 18 for bowel dysfunction.  Also, it may not be appropriate for the more common causes of NBD such as SB and SCI.added 1045-1056

Page 20, line 932.  Many MACEs are placed in the umbilicus., true added 1111-115

In the discussion of MACE, it should be mentioned that many MACE channels are created at the time of surgery for neurogenic bladder when the appendix may also be needed for creation of a Mitrofanoff channel, creating surgical dilemmas on how to best use the appendix in such cases. True, added 1115

Page 20 – Tube cecostomy discussion should mention the disadvantages compared to MACE in that the tube has to be replaced regularly and can be dislodged.  Likewise, the discussion of risk to the VP shunt holds true for MACE procedures as well., true added 1158

Page 20 – the discussion of colostomy should also include that colostomy are sometimes medically necessary for management of sacral decubiti to prevent wound soiling. Added 1156

Page 21 – Bowel resection – Followers of Levitt often recommend bowel resection at the time of MACE creation to allow more complete and faster bowel evacuation.  Certainly, it is controversial topic. Added 1166

Discussion – I felt that the discussion was very pro-TAI even before I read the conflicts.  TAI is good and has had a great effect on NBD management but has many limitations.  Reference 141 demonstrates that is not as effective for continence as antegrade enemas. We agreed with Your observation and modified the text, according to ref 141 Anyway We think it is important to stress TAI is as effective as antegrade enemas which were the previous gold standard before TAI started to gain acceptance and efficacy, and TAI is a no surgical treatment, that can be used anyway as last option.

Reviewer 2 Report

The authors should be applauded for a well written, comprehensive and accurate review of NBD in children and adolescents. Manuscript is well structured and references are generally well placed.

My comments:

Ref (14) should be mentioned after Delphi method and not current literature

Regarding section 3.1.1 cause of NBD, I would consider breaking into NBD secondary to neurological dysfunction/injury (MDS, SCI, TM) and the non-classical NBD, secondary to behavioural and development problems (autism, DS).

Page 3 row 6 details level of MMC, please add reference

Page 4 row 1 delete underscore of A

Page 5 rows 227-231 please clarify if meaning childbirth in adolescents

Page 8 last 2 paragraphs are a repeat of the same sentences

Page 9 bottom – the recto-anal reflex arc (‘cough’ or ‘valsalva’ performed during anorectal manometry with expected increase in anal pressure) is also preserved in UMN injury and lost in LMN. Consider adding.

Page 10 first paragraph: to the best of my knowledge, RAIR is an intrinsic reflec localized to the enteric nervous system. Thus, RAIR is not attenuated in UMN or LMN. Please provide references if otherwise.

Page 15 – biofeedback section. Any role for pelvic floor (non-BF) physiotherapy?

Page 15 line 687 add reference to Bristol stool chart

Page 16 line 764-765 add reference for ‘narrow hyperactive colon’ secondary to phosphate enemas.

Page 18, section 3.2.3.15.1 TENS – there is negative RCT for TENS in FI adults – please cite. (van der Wilt 2017)

Page 19 first paragraph –2 negative RCT trials for SNM in constipation in adults (including Dinning 2016) , currently indicated only for FI. Please add

For management of NBD in SCI – consider to comprehensive review in adults Zhengyan Qi 2018.

Author Response

Rev 2

Dear Editor, Dear Reviewers,

Here enclosed are our corrections according to Your valuable suggestions.

May thanks for Your time and improvements

On Behalf of all Authors

Comments and Suggestions for Authors

The authors should be applauded for a well written, comprehensive and accurate review of NBD in children and adolescents. Manuscript is well structured and references are generally well placed. Thank You very much

Ref (14) should be mentioned after Delphi method and not current literature, done

Regarding section 3.1.1 cause of NBD, I would consider breaking into NBD secondary to neurological dysfunction/injury (MDS, SCI, TM) and the non-classical NBD, secondary to behavioural and development problems (autism, DS). : We respect this suggestion, but We had originally arranged all the conditions in rough order of frequency & relevance according to our experience. We believe that still is logicala s well as Your suggestion, that could suggest to the reader a sense of relative importance of all the conditons that cause NBD. So We would like maintain if this is possible our original classification , thanks

Page 3 row 6 details level of MMC, please add reference, added r 117, ref 16

Page 4 row 1 delete underscore of A, done

Page 5 rows 227-231 please clarify if meaning childbirth in adolescents, done r 264-265

Page 8 last 2 paragraphs are a repeat of the same sentences, true , done

Page 9 bottom – the recto-anal reflex arc (‘cough’ or ‘valsalva’ performed during anorectal manometry with expected increase in anal pressure) is also preserved in UMN injury and lost in LMN. Consider adding.

RAIR is preserved in UMN and in LMN lesions. However in LMN lesions more exaggerated - and this is stated

Lines 434-436 state this and lines 466 to 478

Page 10 first paragraph: to the best of my knowledge, RAIR is an intrinsic reflec localized to the enteric nervous system. Thus, RAIR is not attenuated in UMN or LMN. Please provide references if otherwise.

The reviewer is correct, the RAIR is not attenuated – but rather exaggerated in LMN lesions: (when performing manometry we note, in response to balloon inflation and rectal distention, anal sphincters will relax in an exaggerated response leading to bowel incontinence)

Lines 466-478

A flaccid bowel may follow a lower spinal cord injury. Infra-conal lesions are a consequence of disruption of autonomic motor nerves due to damage to parasympa-thetic cell bodies in the conus medullaris or their axons in the cauda equina. This is characterized by loss of colorectal tone and exaggerated attenuated recto-anal inhibitory reflex (RAIR), resulting in a cyclical pattern of insensate rectal filling and progressive rectal distension eventually leading to fecal incontinence. Furthermore, the incontinence is not helped by a reduction in resting and squeeze anal pressures due to flaccid anal sphincters and laxity of pelvic floor muscles which allows excessive descent of pelvic contents, reducing the anorectal angle and opening the rectal lumen [79]. In a flaccid bowel situation, there is reduced movement in the colon, less peristalsis, and the anal sphincter is more relaxed than normal. This can lead to constipation with frequent leaking of stool. Typically, these patients have no/low resting anal tone, and absence of the anal/anocutaneous and bulbospongiosus/bulbocavernosus reflexes.

Page 15 – biofeedback section. Any role for pelvic floor (non-BF) physiotherapy?

Added r 766-760

Page 15 line 687 add reference to Bristol stool chart, added 137

Page 16 line 764-765 add reference for ‘narrow hyperactive colon’ secondary to phosphate enemas.added

Page 18, section 3.2.3.15.1 TENS – there is negative RCT for TENS in FI adults – please cite. (van der Wilt 2017)

this citation has been included,

3.2.3.10.1. Transcutaneous Electrical Nerve Stimulation

Non-invasive nerve stimulation such as transcutaneous electrical nerve stimula-tion (TENS) is widely used for bowel dysfunction in children: Veiga at al. in 2013 showed a 85.7% improvement in constipation in patients treated with para-sacral TENS [130].

Included 128

Page 19 first paragraph –2 negative RCT trials for SNM in constipation in adults (including Dinning 2016) , currently indicated only for FI. Please add done

For management of NBD in SCI – consider to comprehensive review in adults Zhengyan Qi 2018.

Done ref 41